# Utility and Safety of Bronchoscopic Cryotechniques—A Comprehensive Review

**DOI:** 10.3390/diagnostics13182886

**Published:** 2023-09-08

**Authors:** Shaikh M. Noor Husnain, Abhishek Sarkar, Taha Huseini

**Affiliations:** 1Department of Internal Medicine, Division of Interventional Pulmonary Medicine, Westchester Medical Center, New York, NY 10595, USA; 2Department of Respiratory Medicine, Fiona Stanley Hospital, Perth, WA 6150, Australia

**Keywords:** bronchoscopy, cryobiopsy, interstitial lung disease, lung cancer

## Abstract

Cryosurgical techniques are employed for diagnostic and therapeutic bronchoscopy and serve as important tools for the management of pulmonary diseases. The diagnosis of interstitial lung disease requires multidisciplinary team discussions after a thorough assessment of history, physical exam, computed tomography, and lung-function testing. However, histological diagnosis is required in selected patients. Surgical lung biopsy has been the gold standard but this can be associated with increased morbidity and mortality. Transbronchial lung cryobiopsy is an emerging technique and multiple studies have shown that it has a high diagnostic yield with a good safety profile. There is wide procedural variability and the optimal technique for cryobiopsy is still under investigation. There is emerging data that demonstrate that cryobiopsy is safe and highly accurate in the diagnosis of thoracic malignancies. Furthermore, cryorecanalization procedures are a useful adjunct for the palliation of tumors in patients with central airway obstruction. One should keep in mind that these procedures are not free from complications and should be carried out in a specialized center by a trained and experienced bronchoscopy team. We present a review of the literature on the diagnostic and therapeutic utility of bronchoscopy-guided cryosurgical procedures and their safety profile.

## 1. Introduction

Several bronchoscopic cryosurgical techniques are employed for diagnosis and as an adjunct to treatment for various pulmonary diseases. There has been an exponential increase over the past decade in the data regarding the diagnostic utility, technique, and safety of bronchoscopic cryosurgery with a growing number of interventional pulmonologists and thoracic surgeons performing these procedures.

Bronchoscopy-guided transbronchial cryobiopsy (TBCB) is a useful tool in diagnosing parenchymal lung diseases; it has been proposed as an alternative to surgical lung biopsy (SLB) in the diagnosis of interstitial lung diseases (ILDs) since early 2009 [1]. In selected patients with ILD who require a histological diagnosis, lung biopsies were traditionally obtained by surgery or bronchoscopically via the transbronchial forceps technique. This patient cohort often has multiple comorbidities and compromised ventilatory capacity which puts them at higher risk of having complications including death from invasive surgical procedures [2]. Transbronchial forceps biopsy (TBFB) often lacks sufficient quality due to small size and crush artifacts and has a lower yield in diagnosing ILDs [3,4,5]. TBCB is relatively new and is becoming widespread with a reported accuracy of approximately 80% in diagnosing ILDs [6,7,8,9]. This high accuracy rate, combined with its minimally invasive nature, makes it an attractive alternative to traditional SLB.

A flexible cryoprobe is a useful tool for the diagnosis of lung cancer as well as for therapeutic bronchoscopy, in particular, for cryo-recanalization in patients with central airway obstruction (CAO). Cryobiopsy has a diagnostic accuracy rate of up to 95% for endobronchial tumors [10] and approximately 90% for peripheral pulmonary nodules [11,12], making it a useful tool for diagnosing lung cancer in its early stages, when treatment is most effective.

It is important to note that TBCB or cryosurgery is not suitable for all patients, and alternative diagnostic or therapeutic methods may be preferred. Here, we aim to review the technique, diagnostic and therapeutic utility, and safety of this relatively novel method in different disease modalities.

### Study Selection

We performed a literature review by searching PubMed and Medline databases using the keywords “cryobiopsy” and “interstitial lung disease” OR “lung cancer” OR “central airway obstruction” OR “lung transplant”. Relevant studies that were available in full and peer-reviewed were included; no time frame was set for the inclusion of articles.

## 2. History and Technique

Cryotherapy was first used in the 1800s to stop bleeding and alleviate pain by applying an ice-salt solution to tumors [13]. Babiak et al. initially described the TBCB technique in 2009; they showed larger specimens with a superior diagnostic yield when compared with traditional TBFB [1]. Cryoprobes have existed for decades and have been used in airway tumor debulking, removal of foreign bodies, and blood clots [14].

Cryosurgical equipment works via the Joule–Thomson effect which shows that compressed gas (e.g., nitrous oxide or carbon dioxide) at rapid flows can generate extremely low temperatures. Cryoprobes are insulated catheters with blunt metal tips which cool rapidly to extremely cold temperatures. The flexible cryoprobes are available in various designs and sizes. The patterns of TBCB vary depending on individual skills and expertise. Cryobiopsy can be performed via rigid or flexible bronchoscopy; the advantage of rigid bronchoscopy is that of isolation and ventilation of the contralateral airway and rapid introduction and removal of instruments to deal with procedural complications (e.g., bleeding). Figure 1 shows an example of a TBCB procedure. The cryoprobe is advanced through the working channel of a flexible bronchoscope into the target peripheral lung segment; this is then activated, resulting in a rapid freeze of the tip which adheres to the lung tissue. Traditionally, the bronchoscope and the cryoprobe are removed en bloc, given that the biopsy specimen is generally larger than the working channel of the flexible bronchoscope. Once the specimen is released from the cryoprobe, it is removed and the bronchoscope is re-inserted into the airway. However, with the advent of newer and thinner 1.1 mm cryoprobes with an oversheath, TBCB can be performed without having to remove the bronchoscope from the airway, offering a good safety profile [15].

## 3. Diagnostic and Therapeutic Utility

### 3.1. Interstitial Lung Disease

ILD is a broad term encircling a group of lung diseases that are often difficult to diagnose. In recent years, multidisciplinary team discussions (MDD) have overtaken the previous gold standard of SLB. They are a vital process and remain a critical step in the diagnostic pathway, improving diagnostic confidence [16]. Usually, a consensus diagnosis is made by MDD after reviewing clinical and radiological data but occasionally histopathological specimens are required when evaluating these patients [16,17]. Radiological guidelines report that a diagnosis of idiopathic pulmonary fibrosis (IPF) can be made without an SLB in selected cases when computed tomography (CT) shows a probable usual interstitial pneumonia (UIP) pattern [18]. However, for patients with CT features indeterminate for UIP or when the CT is suggestive of an alternate pathology and the ILD remains unclassifiable, diagnostic tools, and techniques to obtain tissue specimens may be needed for definite diagnosis. An ideal test is one which achieves the principal goal of adequate diagnostic tissue, is minimally invasive, cost-effective, can be carried out as an outpatient, and has a good safety profile.

Table 1 summarizes major studies describing TBCB as a diagnostic tool for ILD. It has already been established that TBCB allows for the procurement of larger pieces of tissue without crush artifact when compared to TBFB [5]. In a two-center prospective study by Romagnoli et al., Cryo-PID Study, 21 patients’ TBCB samples were analyzed with a diagnostic yield of 81%. However, there was a poor correlation with the results of the SLB (diagnostic concordance between techniques was 38%, kappa 0.22) [19]. In another notable prospective, multicenter, comparative study (COLDICE), diagnostic agreement between TBCB and SLB was conducted across nine Australian tertiary hospitals. Both techniques showed high performance (93% for TBCB, 98% for SLB). Contrary to the Cryo-PID study, concordance between the two techniques to establish a histological diagnosis was 70.8% (kappa 0.7), and concordance between the two techniques by MDD was 76.9% (kappa 0.62) [7]. Both of these major prospective studies showed a good safety profile with no major life-threatening bleeding events and a low rate of pneumothorax of reportedly 2 out of 21 and 1 out of 65 cases for CryoPID and COLDICE, respectively.

The independent diagnostic yield of TBCB for ILD is difficult to calculate, given there is no standard definition of a “diagnostic sample”. However, several meta-analyses and systematic reviews have analyzed it. A systematic review by Sharp et al. which included 704 patients from 11 studies concluded a diagnostic yield of 84.4% (95% confidence interval [CI], 75.9–91.4%) [20]. Similarly, a meta-analysis by Sethi et al. in 2019 analyzing 27 studies including 1443 patients, reported an overall diagnostic yield of 72.9% [95% CI 67.9–77.7%] [21]. Johannson et al. looked at 11 studies that included 731 patients and showed that that the diagnostic yield of TBCB ranged from 74 to 98%, with a pooled estimate of 79% [22]. A meta-analysis from Ravaglia et al. of 15 studies including 781 patients revealed an overall diagnostic yield of 81%. Furthermore, they retrospectively analyzed 297 patients who underwent TBCB and 150 patients who underwent SLB for diagnosis of ILD, median time of hospitalization was 2.6 days with TBCB vs. 6.1 days with SLB. Mortality in their cohort was 0.3% for TBCB vs. 2.7% with SLB [23]. A cross-sectional study by Tomassetti et al. [24] observed a major increase of 34% in diagnostic confidence after the addition of TBCB (from 29% to 63%, *p*-value = 0.0003), similar to SLB. In cases where there is low confidence for a diagnosis of IPF, the addition of histological data from TBCB had the most significant impact. 

TBFB and endobronchial ultrasound (EBUS)-guided transbronchial needle aspiration (TBNA) of lymph nodes has a relatively high yield in patients with suspected sarcoidosis. However, TBCB may be a useful tool in selected cases of suspected sarcoidosis where TBFB has been inconclusive [25,26].

International clinical practice guidelines, published in 2018, recommend SLB for obtaining a histopathological diagnosis and do not include TBCB in the diagnostic algorithm for ILD [27]. However, the role of TBCB was not discouraged in the same guidelines. Since there is emerging evidence in support of TBCB as an alternative to SLB, these guidelines may change shortly. In 2020, the American College of Chest Physicians (ACCP) Guidelines and Expert Panel Report did conclude that TBCB is a reasonable alternative to SLB for providing histology for ILD MDD [28]. 

#### Pathological Considerations

TBCB yields specimens sized between forceps and surgical biopsies, hence balancing the increased diagnostic yield of the latter with the safety of the former [29,30]. The histopathologic assessment of cryobiopsy-obtained tissue samples undergoes the same pathological assessment as those obtained via traditional methods, like SLB or TBFB [29,31]. This includes processing, fixing, and staining with hematoxylin-eosin, elastica van Gieson, and Prussian blue along with any specific antibodies vital to immunohistologic diagnosis [32,33,34,35]. Analysis usually begins with the microscopic examination at low power to determine the presence of a histological pattern, then the pattern and differential diagnoses are correlated with the clinical and radiologic findings to complete a multidisciplinary approach which has been shown to increase the diagnostic yield [31,34].

When analyzing a cryobiopsy specimen, consideration must first be given to the artifacts or inclusions that may deter a pathologist from making a diagnosis. Cryobiopsy has been shown to have fewer frozen artifacts when compared to cryostat sections, and while they may lack enough detail in the nuclear and cytoplasmic features, this does not have a significant effect on pathologic interpretation [29]. Further, the cryobiopsy technique minimized the inclusion of crush artifacts usually seen in samples obtained via conventional TBFB [30,34,35]. The trauma to the tissue that occurs while obtaining a biopsy may lead to the inclusion of adjacent tissue such as bronchial epithelium, visceral pleura, parietal pleura, intra-alveolar material, skeletal muscle, vessels, or blood from procedure-induced hemorrhage [24,29,34,36]. Hemorrhagic artifacts are often focal findings that do not interfere with pathologic interpretation, but if the artifact is extensive, it may inadvertently lead to the suspicion of an acute lung injury [34]. Depending on the angle of the biopsy, airways may take up a large portion of the tissue sample, especially in patients with obstructive airway diseases, and thus multiple biopsies (3–5 samples per lobe) with large surface areas should be taken [29,34]. Care should be taken to ensure that the bronchoscope fully passes the smallest subsegment and is placed in the outer one-third of the lungs in order to obtain alveolar tissue and minimize the bronchial wall [34]. Common causes of a nondiagnostic sample obtained via cryobiopsy are either the presence of bronchial wall tissue or of normal lung parenchymal tissue, both often due to procedural error in sampling, likely stemming from the lack of procedural standardization [31,34,37].

When analyzing for histological patterns, the smaller size of the sample may be a hindrance but typical patterns are clearly visible as long as the samples obtained are at least 5 mm [29,31,33,38]. The diagnosis of UIP can be made via the identification of patterns like honeycombing, patchy fibrosis, and fibroblastic foci, the visualization of which can be impaired by the presence of crush artifact or small sample size when obtained by TBFB [29,30,33,34]. A study by Casoni et al. showed that, of all the instances of TBCB in UIP and IPF, only 8.7% were considered non-diagnostic [30]. A similar diagnostic rate was claimed by Hagmeyer et al. who showed that 91% of the TBCB samples were indicative of the initial diagnosis with 72% requiring no further diagnostic intervention [32].

Similarly, patterns of fibrosis consistent with nonspecific interstitial pneumonia (NSIP) were easily identified via cryobiopsy samples and demonstrated a high diagnostic yield [34]. The same can be said for non-necrotizing granulomas, Langerhans histiocytes, and stellate lesions in samples obtained for suspicion of sarcoidosis [29,34,39]. When investigating eosinophilic pneumonia, cryobiopsy was able to produce a sample lacking any crush artifacts that allowed the identification of lymphoid follicles, septal widening, giant cells, and eosinophilic nests in levels rivaling samples obtained via surgical biopsy [29,32,34]. Samples obtained for the suspicion of subacute hypersensitivity pneumonitis were able to clearly demonstrate bronchiolocentric accentuation, non-necrotizing granulomas, and septal chronic inflammation at levels high enough to make a confident and definitive diagnosis [34].

With the above considerations, cryobiopsy has been shown to have a diagnostic yield in the close to 90% range, approaching the confidence of surgically obtained biopsies while minimizing complications [8,33,34,36,37].

### 3.2. Thoracic Malignancy

Lung cancer is one of the leading causes of cancer-related deaths worldwide [40]. With the advent of targeted therapies especially for non-small cell lung cancer (NSCLC), survival has improved [41]. However, this is an aggressive malignancy, and timely diagnosis and initiation of treatment are pivotal. Obtaining adequate tissue samples is key for the diagnosis and subtyping of cancers to guide further management. Inadequate biopsy specimens may require repeat procedures and delay management. This can compromise treatment and patients may miss the window of opportunity for cure [42]. Furthermore, there are extra costs involved in repeat procedures which adds a financial burden on the health system.

Endoscopic procedures are minimally invasive and are the preferred initial investigation for the diagnosis of lung cancer. EBUS-guided TBNA is an excellent minimally invasive tool for obtaining a cytologic diagnosis for mediastinal lesions. The literature reports a diagnostic yield of standard linear EBUS of around 89–93.5% [43,44,45]. However, in the era of targeted therapies especially for NSCLC, adequate tissue is essential for molecular analysis and ancillary testing. Furthermore, rare tumors or hematological malignancies such as lymphoma may require a core biopsy rather than a fine needle aspirate to obtain a larger tissue sample and a histological diagnosis. Zhang et al. showed that cryo-biopsy may be a useful adjunct to diagnostic bronchoscopy for mediastinal lesions. They safely carried out a randomized trial of EBUS-guided TBNA and TBCB for mediastinal lesions in 196 patients. They reported improved diagnostic yield with TBCB as compared to conventional TBNA (91.8% vs. 79.9%, *p*-value = 0.001) [46].

TBCB is emerging as an adjuvant tool to radial EBUS (r-EBUS) in diagnosing peripheral pulmonary lesions (PPLs). A recent large observational study included 1024 patients with PPLs ≥ 2 cm in size. PPLs were localized with r-EBUS, fluoroscopy, or both and TBCB was performed. TBCB appeared safe with a relatively low complication rate; significant bleeding occurred in 3.5% of cases and pneumothorax requiring drainage occurred in 6.6% of patients. The diagnostic yield of TBCB was 91%, [11]; this is a great improvement on the reported yield of around 70% with standard biopsies via navigational techniques such as r-EBUS and electromagnetic navigation (EMN) [47,48]. Torky et al. showed that r-EBUS-guided TBCB is a safe and effective technique with a higher diagnostic accuracy and better-quality samples compared to forceps biopsies [49]. Nasu et al. [50] compared TBCB and TBFB under r-EBUS guidance for 53 PPLs and showed no significant difference in diagnostic yield (86.8 vs. 81.1%, *p*-value = 0.6). However, on further analysis, all tests on patients who tested positive for TBFB and negative for TBCB were performed without a guide sheath (GS) and TBCB carried out with GS had a better diagnostic yield than without GS. Oberg et al. [51] retrospectively analyzed 112 patients with 120 PPLs who had a biopsy via robotic bronchoscopy. Patients had TBNA followed by TBFB and a TBCB with a 1.1 mm cryoprobe. Overall diagnostic yield was 90%; out of these, TBNA was positive at 31.5%, TBFB at 77.8%, and TBCB at 97.2%. TBCB yielded a diagnosis exclusively in 18% of the cases and samples were adequate for molecular analysis. Hetzel et al. demonstrated that obtaining an endobronchial cryobiopsy had a superior diagnostic yield when compared to traditional forceps biopsy (95 vs. 85.1%, *p*-value < 0.001) [10]. Table 2 summarizes the above studies describing TBCB as a diagnostic tool for various thoracic malignancies.

An adequate sample size is often required to perform molecular testing and cryobiopsy appears feasible for diagnostic assessment in lung cancer. Nishida et al. obtained conventional scalpel biopsy and cryobiopsy in 43 surgically resected primary lung tumors. Samples were prepared for immunohistochemical stains and a high concordance between the two sampling techniques was observed for certain tumor markers including programmed cell death ligand 1 (PD-L1) [52]. Furthermore, bronchoscopic cryobiopsy can increase the detection of epidermal growth factor receptor (EGFR) mutation in comparison to other tissue sampling techniques [53]. Given the cytotoxic effects of extreme cold on living tissue, cryotherapy can also be used for the treatment of endobronchial tumors, in particular, in-situ carcinomas, early-stage lung cancers, or stalks of resected tumors [54]. Lastly, due to its well-preserved specimens, cryobiopsy has been shown to be diagnostic for malignant pleural mesothelioma when conventional methods have previously failed [35].

### 3.3. Central Airway Obstruction

CAO from a tumor is a life-threatening emergency and can result in death from atelectasis, respiratory failure, or pneumonia. Tumor debulking to relieve CAO can be achieved by laser, argon plasma coagulation (APC), mechanical endoscopic debridement, or cryorecanalization. There is a risk of airway fire with APC or laser which requires a reduction in oxygenation whilst these modalities are being applied. This can be challenging in patients with CAO who often present with respiratory failure and have compromised ventilatory capacity. Cryotherapy offers the advantage of maintaining oxygenation during the procedure, given that this works via a freezing effect, hence no risk of airway fire. Furthermore, cryotherapy equipment is less expensive than laser, making it a cost-efficient method for palliation of CAO. A cryoprobe can be introduced via flexible or rigid bronchoscopy into the tumor which is frozen and then extracted piecemeal. Schumann et al. reported successful cryorecanalization and rapid improvement from CAO in 91.1% of 225 patients. Moderate bleeding occurred only in 8% of patients which was controlled with APC or bronchial blocker [55]. A systematic review of 16 studies on endoscopic cryotherapy showed an approximately 80% success rate from cryorecanalization of CAO. In this review, most studies looked at patients with advanced inoperable lung cancer and the cryotherapy resulted in a statistically significant improvement in dyspnea, stridor, lung function, and quality of life scores [56].

### 3.4. Blood Clot and Foreign Body Retrieval

Massive pulmonary hemorrhage, regardless of the etiology, can cause tracheobronchial obstruction from blood clots and life-threatening ventilatory failure. Bronchoscopic cryoextraction can facilitate the removal of blood clots and mucus plugs [57]. Small case series have shown that this can be achieved via flexible or rigid bronchoscopy to relieve tracheobronchial obstruction and provide improvement in ventilation [57,58,59,60]. Similarly, the cryo-adhesive effect on objects with water content can facilitate the extraction of foreign bodies from airways and its use has been described in multiple case reports [57,61,62]. Foreign bodies without enough water content can be extracted by spraying saline on the object followed by immediate freezing by the cryoprobe.

### 3.5. Lung Transplant

In recent years, the number of lung transplants has increased worldwide. However, graft dysfunction rates in lung transplantation recipients continue to be high with post-transplant rejection commonly seen as one of the major contributors [63,64]. The International Society for Heart and Lung Transplantation reports 28% of lung transplant recipients experience at least one episode of acute rejection in the first year following transplantation [65]. For this reason, lung transplant recipients undergo routine surveillance lung biopsies. Conventionally, TBFB has been the main technique to establish the presence of lung allograft rejection (AR) but there has been a trend towards TBCB in recent years.

Frutcher et al. published the first case series of 40 lung-transplant recipients in 2013, comparing TBCB with traditional TBFB [66]. The primary outcome focuses were procedure characteristics, complications, and diagnostic yield. The mean diameter of TBCB was 10 mm^2^ compared with 2 mm using TBFB (*p*-value < 0.05) which translated to a significant increase in the percentage of alveolar tissue (65 vs. 34%, respectively, *p*-value < 0.05), clear histological detection of acute AR (*n* = 4), pneumonitis (*n* = 3), diffuse alveolar damage (*n* = 1) and confident exclusion of acute AR, infection or pneumonitis (*n* = 32). Fluoroscopy time was significantly shorter with TBCB compared with TBFB (25 vs. 90 s, respectively, *p*-value < 0.05) and there were no major complications observed. Mohamed et al. showed an improved diagnostic rate of acute AR (100 vs. 83%, *p*-value < 0.001) and chronic AR (85 vs. 64%, *p*-value < 0.001) with TBCB compared with TBFB [67].

In another retrospective analysis of 402 patients, TBFB and TBCB specimens from lung-transplant recipients were reviewed; acute AR was diagnosed in 21.9% of the TBCB group vs. 14.9% in the TBFB group (*p*-value = 0.09). Larger specimen sizes were obtained from TBCB when compared with TBFB (16.6 vs. 6.6 mm^2^, *p*-value < 0.001) with less crush and bleeding artifact (*p*-value < 0.001) [68]. Similarly, Yarmus et al. also showed larger specimens were obtained with TBCB from 21 procedures in 17 patients with no significant procedural complications; all patients were discharged on the same day of the procedure [69].

Although the data available is limited, the above studies show that TBCB can be safely performed in lung-transplantation recipients and provides larger specimens with increased diagnostic yield.

## 4. Complications

Generally, postprocedural pneumothorax and bleeding are the most common procedural complications associated with TBCB [70]. The reported rate of pneumothorax varies amongst different publications and is anywhere from 1–30% [5,32,69,71]. A meta-analysis from Iftikhar et al. [70] reported an average of 9.5% postprocedural pneumothorax and a 0.7% 30-to-60-day mortality compared with 1.8% with SLB [23]. Although the incidence of pneumothorax reported in this meta-analysis is higher than that of traditional TBFB, given the higher diagnostic yield and overall safety profile compared with SLB, the authors suggested that TBCB can be a reasonable first line procedure in clinical practice for patients who require a biopsy procedure. Other major risk factors described for pneumothorax are UIP histology, fibrotic reticulation on CT scan, and biopsies taken close to the pleura [23,30].

Another common complication with TBCB is bleeding which is generally controlled successfully via endoscopic techniques [23,30,32,69,72,73]. However, severe bleeding can be life-threatening and requires a trained endoscopy team to deal with this complication. DiBardino et al. showed a 12% rate of serious hemorrhage including one life-threatening bleed in 25 patients following TBCB [74].

## 5. Contraindications

Contraindications in relation to the complication of severe bleeding include the presence of thrombocytopenia (platelet count < 50 × 109/L), use of anticoagulant medications and antiplatelet agents such as clopidogrel; the use of aspirin is generally not considered a contraindication. Severe pulmonary hypertension with a pulmonary artery systolic pressure (PASP) of > 50 mmHg on transthoracic echocardiogram (TTE) is another relative contraindication to TBCB [33]. In some studies, patients with PASP > 40 mmHg on TTE have been excluded as there is a higher tendency to bleed in patients with pulmonary hypertension [1,23,73].

There are also data suggesting the exacerbation of existing ILD leads to mortality [30,32], hence acute deterioration in respiratory status should be considered a relative contraindication. Similarly, severe impairment in lung function has also been shown to be related to increased complications such as pneumothorax and it is suggested that the diffusing capacity of the lung for carbon monoxide (DLCO) <35% or forced vital capacity (FVC) <50% should be considered as relative contraindications [33]. Although complications from TBCB can happen independent of lung function [23], most of the data are variable and derived from earlier studies in relation to SLB and not TBCB. Significant hypoxemia defined as the partial pressure of oxygen on arterial blood gas of <55 mm Hg on room air or while receiving supplemental oxygen of ≥2 L/min has also been mentioned as a relative contraindication [3,30,73].

In general, bronchoscopic cryosurgery requires procedural sedation or general anesthesia. Therefore, higher peri-operative patient-specific risk factors should be considered as a contraindication. Given the varied criteria for contraindications amongst all these studies, patient selection for this procedure should be individualized after a thorough assessment of the risk-to-benefit ratio.

### Future Directions

Future perspectives include performing TBCB for investigation of ILD under robotic, navigational bronchoscopy or r-EBUS guidance with the use of real-time CT or augmented fluoroscopy to target diseased segments. The advantage of r-EBUS-guided cryobiopsy is that adjacent blood vessels can be visualized and the risk of bleeding can be minimized by targeting the biopsy with no large adjacent blood vessels [75]. Furthermore, new tools are being developed such as an outside-the-scope catheter system with which the cryoprobe is attached outside the bronchoscope via an external working channel and can be removed free from the bronchoscope providing real-time visualization of the airway. This may hold promise for increasing procedural safety and minimizing bleeding-related complications [76]. With advancements in cancer therapies which ideally require a maximum volume of tissue for genotyping and phenotyping, TBCB may be considered the first-line tool for the diagnosis of lung cancer.

## 6. Conclusions

Cryosurgical procedures are rapidly emerging with high diagnostic yield and good therapeutic effects. However, these procedures are complex and not free from complications. The proceduralist and the team assisting should be trained and competent in dealing with potentially life-threatening complications. In this era of modern interventional pulmonology, cryosurgical techniques should be part of the training and should be performed by skilled and experienced proceduralists working in specialized centers [77].

## Figures and Tables

**Figure 1 diagnostics-13-02886-f001:**
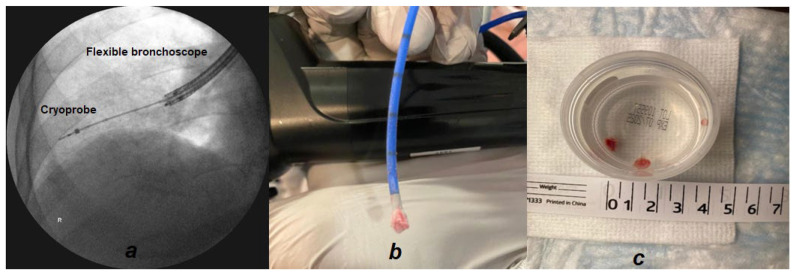
Transbronchial cryobiopsy procedure. (**a**) Fluoroscopic view of transbronchial lung cryobiopsy in the right lower lobe of a patient with intersitial lung disease. (**b**) Bronchoscope and cryoprobe removed en bloc with frozen specimen at the tip of the cryoprobe. (**c**) Cryobiopsy specimens released from the cryoprobe into specimen container.

**Table 1 diagnostics-13-02886-t001:** Study characteristics describing TBCB as a tool for diagnosing ILD.

Author	Year	Study Design	No. of Patients	Comparison	TBCB Size	Diagnostic Yield	Diagnostic Agreement with SLB	Diagnostic Confidnce after MDD	Complications	Comments
Romagnoli et al. [19]	2019	Prospective	21	TBCB vs. SLB	7 mm (5–8 mm) ^a^	80.95%	38%, kappa 0.22	48%, kappa 0.31	Pneumothorax: 9.5%	TBCB vs. SLB for diagnosis of ILD. 5–6 s freeze time with TBCB
Troy et al. [7]	2020	Prospective	65	TBCB vs. SLB	7.1 mm ±1.9 ^b^	90.90%	70.8%, kappa 0.7	76.9%, kappa 0.62	Pneumothorax: 1.5%	Study investigating diagnostic agreement between TBCB and SLB for ILD.
Sharp et al. [20]	2017	Systematic review	704	TBCB vs. TBFB vs. SLB	NA	84.40 vs. 64.3 vs. 91.1%	NA	NA	Pneumothorax: 10%	Systematic review of 11 studies looking at relative diagnostic yields and safety for
									Moderate bleeding: 20.99%	TBCB, TBFB and SLB.
									Mortality: 0.5%	
Sethi et al. [21]	2019	Systematic review	1143	NA	Mean: 23.4 mm (95% CI, 9.6–37.3 mm)	72.90%	NA	NA	Pneumothorax: 9.4%	Systematic review of 27 studies assessing diagnostic yield
									Significant bleeding: 14.2%	and safety profile of TBCB in ILD. Variable freeze times.
									Mortality: 0.3%	
Johannson et al. [22]	2016	Systematic review	731	NA	6.6–64.2 mm^2^	74–98%	NA	NA	Pneumothorax: 12%	Systematic review of 11 studies looking at diagnostic accuracy of TBCB for ILD.
						(pooled estimate 83%)			Moderate/severe bleeding: 39%	
Ravaglia et al. [23]	2016	Retrospective	447	TBCB vs. SLB	NA	82.8 vs. 98.7%	NA	NA	Pneumothorax: 20.2% Mortality: 0.3%	Retrospective analysis of 297 patients with TBCB and 150 patients with SLB for ILD.
Tomassetti et al. [24]	2015	Cross-sectional study	117	TBCB vs. SLB	NA	NA	NA	63% (TBCB) vs. 65% (SLB)	Pneumothorax: 33%	Impact of TBCB vs. SLB on diagnostic confidence during MDD
									Mortality: 1.7%	Rigid bronchoscopy with propofol and remifentanyl used as anesthesia for TBCB.

Abbreviations: TBCB, Transbronchial Cryobiopsy; ILD, Interstitial Lung Disease; SLB, Surgical Lung Biopsy; MDD, Multidisciplinary Team Discussions; TBFB, Transbronchial Forceps Biopsy; NA, Not Available/Not Applicable; CI, Confidence Interval. ^a^ Values are expressed as median (interquartile range). ^b^ Values are expressed as mean ± standard deviation.

**Table 2 diagnostics-13-02886-t002:** Study characteristics describing TBCB as a tool for diagnosing thoracic lesions.

Author	Year	Study Design	No. of Patients	Comparison	Biopsy Size	Diagnostic Yield	Complications	Comments
Zhang et al. [46]	2021	RCT	197	TBCB vs. TBNA	10.7 mm^2^	91.8 vs. 79.9%, *p*-value *=* 0.001	No significant difference in majorcomplications requiring intervention.	EBUS guided TBCB vs. TBNA for mediastinal lesions both malignant and benign etiologies.
Herth et al. [11]	2021	Retrospective	1024	NA	NA	91%	Pneumothorax requiring intervention 6.3%.Grade 3 bleeding (3.5%).	TBCB for diagnosis of PPLs under r-EBUS and/or fluoroscopy guidance.
Torky et al. [49]	2021	Prospective	60	TBCB vs. TBFB	7.3 ± 2.1 mm vs.3.9 ± 1.6 mm ^a^	76.7 vs. 69.8%	1.7% significant bleeding.	TBCB vs. TBFB for diagnosis of PPLs
							1.7% pneumothorax.	under r-EBUS guidance.
							1.7% hypoxemia.	
Nasu et al. [50]	2019	Retrospective	53	TBCB vs. TBFB	14.1 mm^2^ (range 3.67–40.7 mm^2^) vs.	86.8 vs. 81.1%, *p*-value *=* 0.6	No severe bleeding	TBCB vs. TBFB for diagnosis of PPLs
					2.62 mm^2^ (range 0.737–10 mm^2^),		No pneumothorax	under r-EBUS guidance.
					*p* < 0.001			
Oberg et al. [51]	2022	Retrospective	112	TBNA vs. TFBF vs. TBCB	NA	31.5 vs. 77.8% vs. 97.2%	5.4% pneumothorax	TBNA vs. TBFB vs. TBCB for diagnosis of
							2.7% minor bleeding	PPLs via robotic bronchoscopy under r-EBUS
								and fluoroscopy guidance.
Hetzel et al. [10]	2012	RCT	593	EBCB vs. EBFB	NA	95.2 vs. 82.2%, *p* > 0.001	No significant difference in major bleeding requiring intervention 18.2% vs. 17.8%.	RCT comparing cryobiopsy vs. traditional forceps biopsy for endobronchial lesions.

Abbreviations: TBCB, Transbronchial Cryobiopsy; RCT, Randomized Control Trial; TBNA, Transbronchial Needle Aspirate; EBUS, Endobronchial Ultrasound; NA, Not Available; r-EBUS, Radial-Endobronchial Ultrasound; PPLs, Peripheral Pulmonary Lesions; TBFB, Transbronchial Forceps Biopsy; EBCB, Endobronchial Cryobiopsy; EBFB, Endobronchial Forceps Biopsy. ^a^ Values are expressed as mean ± standard deviation.

## Data Availability

Not applicable.

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
