# Peer review of "Utility and Safety of Bronchoscopic Cryotechniques—A Comprehensive Review"

_diagnostics, 2023, doi:10.3390/diagnostics13182886_

Round 1

Reviewer 1 Report

Contributions of the work is missing

Extensive review has to be carried out referring latest papers

A comparative analysis as to be performed

The contributions of the work should be listed Authors are suggestion to carry thorough literature review of latest papers and perform a comparative analysis of the existing works

accept

Author Response

Contributions of the work is missing:
Thank you, author contribution has been updated in the manuscript. This is present at the end after the conclusion.

Extensive review has to be carried out referring latest papers: 
Thank you, we have carried extensive literature search and included more relevant papers in the revised version. Updated paragraphs have been highlighted in red e.g. line 134-137, 243-252, 338-341 and 377-387

A comparative analysis as to be performed:
Thank you, table 1 and 2 have been added comparing cryobiopsy with other modalities and summarizing latest literature. 

Reviewer 2 Report

Dear authors,

thank you for submitting the review.

Althougt the subject is of importance for pneumologists and thoracic surgeons there are some difficulties: Due to missing structure the text is very difficult to read. Please use diagrams/ charts for visualisation of the essentials. Indications, complications and contraindications should cleary be pointed out.

The systematic of the review process is not explained.

Author Response

Althougt the subject is of importance for pneumologists and thoracic surgeons there are some difficulties: Due to missing structure the text is very difficult to read. Please use diagrams/ charts for visualisation of the essentials. Indications, complications and contraindications should cleary be pointed out.

Thank you, we have included 1 figure and two tables and have further restructured headings with separate paragraphs for complications and contraindications in the revised version which should make it easier for the reader. These have been highlighted in red. 

The systematic of the review process is not explained.

Thank you, we have updated a paragraph line 64-68 with study selection explaining our systematics. We have written this up as a review article and therefore, some systematics which are usually present in a meta-analysis / systematic review are not applicable here. We hope this paragraph briefly explains the systematics of this review article. 

Reviewer 3 Report

I read with great interest the work of Husnain and Huseini, named “Utility and Safety of Bronchoscopic Cryotechniques; A Comprehensive Review". This review is very interesting, addressing a recent issue as cryotherapy in lung

English language should be revised, due to mistakes and imprecisions.

I have some comments:

-In treating central airway obstruction, why should cryotherapy be preferred to laser o argon plasma coagulation? Please, argue it in the paper

- you could add photos concerning the instrumentation and, if available, some examples of the application of this technique 

there are errors and imprecisions in the text

Author Response

English language should be revised, due to mistakes and imprecisions.

Thank you, we have updated all grammatical and vocabulary errors. Furthermore, author 2 is a native English speaker and has independently read and corrected all language based imprecisions. 

-In treating central airway obstruction, why should cryotherapy be preferred to laser o argon plasma coagulation? Please, argue it in the paper

Thank you for raising an important issue, this has been updated in the manuscript. Hope it answers the question. See line 273-279 under heading of central airway obstruction. 

you could add photos concerning the instrumentation and, if available, some examples of the application of this technique 

Thank you, figure 1 has been added describing a transbronchial lung cryobiopsy procedure. 

Reviewer 4 Report

Attached is my comments to the authors.

Author Response

Comment 1:

Thank you, Table 1 and Table 2 have been added summarizing comparisons of major studies addressing the role of cryobiopsy in ILD and lung cancer. 

Comment 2:

Thank you for raising an important point, a paragraph future directions have been updated line 376-387, this has been highlighted in red. 

Round 2

Reviewer 2 Report

Dear authors,

thank you for the reviesed paper with clearance of selstion data and clear definition of complication rates and future aspects.